# Replication of a Culturally Tailored Tobacco Cessation Intervention for Arab American Men in North Carolina: An Exploratory Pilot Study

**DOI:** 10.3390/ijerph22091453

**Published:** 2025-09-19

**Authors:** Dana El Hajj, Linda Haddad, Anastasiya Ferrell

**Affiliations:** 1Loewenberg College of Nursing, University of Memphis, Memphis, TN 38152, USA; lhaddad@memphis.edu; 2School of Nursing, University of North Carolina Wilmington, Wilmington, NC 28403, USA; ferrella@uncw.edu

**Keywords:** smoking, motivational interview, Arab culture, Arab American, nicotine replacement therapy

## Abstract

(1) Introduction: Arab American (ArA) men have higher smoking rates than the general population, driven by cultural norms. Culturally tailored interventions that incorporate ArA cultural, linguistic, and social contexts are essential for addressing tobacco use and promoting health equity. This study aimed to evaluate a culturally tailored smoking cessation intervention for ArA men living in North Carolina. (2) Methods: This pilot study employed a one-group pre- and post-test design to evaluate program effectiveness within financial and time constraints. The participants completed questionnaires and Carbon monoxide measurements and were provided with Nicotine Replacement therapy. (3) Results: The study found that participants experienced anxiety and stress when delaying their first morning cigarette, which hindered cessation. Although smoking was reduced, relapse was common, highlighting the need for personalized support, especially for those with higher nicotine dependence. While telephone Motivational Interviewing helped reduce anxiety, it was insufficient for complete cessation, underscoring the need for tailored approaches addressing both psychological and physical factors. (4) Conclusions: The study suggests that culturally tailored telephone counseling did not show promise as a smoking cessation strategy for Arab Americans in North Carolina due to low participation. The sample size is really too small to test the efficacy of the intervention itself. It seems to have been more successful in another state. Future efforts should address cultural factors, emerging nicotine products, and expanded research. The project is significant for addressing health disparities among Arab Americans by integrating culturally relevant smoking cessation strategies with evidence-based methods like Nicotine Replacement Therapy.

## 1. Introduction

With increased ethnic diversity in the United States, cultural origins strongly influence views on tobacco [1], especially among Arab Americans (ArA)—an ethnic minority group with roots in 22 Arab countries [2]. Arab Americans, who number nearly 3.7 million [2] and are one of the fastest-growing immigrant groups in the country, tend to preserve cultural traditions after immigrating, including practices tied to language, religion, and family dynamics. However, research into tobacco use in this population is limited. Prevalence of smoking among ArA men ranged from 26% to 45% compared to 24% among the U.S. male population [3,4]. In many Middle Eastern countries, smoking is a common cultural practice among men [5]. Consequently, secondhand smoke exposure is a significant risk for members of ArA households, which frequently include extended family members.

Although a few smoking cessation programs have been developed for ArAs [6], mainstream cessation resources often fall short, as language barriers and limited English fluency [7] remain obstacles for this population. This language gap persists because many ArA continue using Arabic for religious and cultural practices, such as worship in mosques or Orthodox churches. ArA men also face additional challenges accessing cessation programs, including demanding work schedules due to underemployment and experiences of social exclusion, often due to stereotypes and misunderstandings linked to global Middle Eastern issues [8,9]. Without culturally relevant resources, smoking behaviors are likely to persist across ArA communities.

Researchers argue for culturally adapted tobacco interventions that address health inequities [6]. While “targeted” interventions adapt general programs for minority use, “tailored” interventions fully incorporate the cultural and social norms of specific groups, making them more effective. This cultural tailoring includes “surface structure”—using culturally aligned staff, language, and symbols—and “deep structure,” which integrates a group’s cultural, historical, and psychological background into program design.

This study aims to evaluate *Sehatack*, a culturally and linguistically tailored smoking cessation program for Arab American (ArA) men in North Carolina. Despite high smoking rates within this population, limited research has explored the cultural factors influencing tobacco use, and Arabic-language cessation resources remain scarce. Addressing these gaps, this study replicates an earlier intervention conducted in Florida [10], using a one-group pre-test/post-test design with 79 participants. Findings indicated increased interest in quitting and a significant reduction in daily cigarette use, suggesting that *Sehatack* is a promising approach for this underserved group. Strong community engagement contributed to the program’s success, though larger trials are needed to confirm its effectiveness.

## 2. Materials and Methods

This study had a one-group pre- and post-test design (see Figure 1) to evaluate the efficacy of the program and feasibly measure change in participants’ behavior within the financial and time constraints. Recruitment of study participants occurred in places of faith and community centers between January 2020 and May 2022. The “Sehatack” intervention included weekly phone counseling and workbooks completed in Arabic within a four-week period.

Following the clinical guidelines for combined treatment [10], a four-week supply of nicotine replacement therapy (patches, lozenges, gum) was provided based on subjects’ nicotine dependence. Questionnaires and Carbon Monoxide (CO) measurements were administered at baseline and during follow-ups to track participants’ progress. Participants received a USD 10 gift card for each follow-up. CO measurements were completed using the iCOquit Smokerlyzers, which were purchased for the participants and were provided for them to keep for their individual use. The present research design can be conceptualized via the stages of change in the Transtheoretical Model (TTM) [11,12] as shown in Figure 2. The “Sehatack” intervention was developed based on Motivational Interviewing (MI) related to the TTM. At the time of recruitment (Baseline/Week 0), potential study participants were not fully ready to quit smoking and might not have even thought of this behavior change, which put them in the precontemplation stage. In Week 1 of the intervention, study participants moved into the contemplation stage when they started Workbook #1 of the “Sehatack” intervention, as it encouraged them to begin thinking about the costs and benefits of smoking. In Week 2, participants reviewed Workbook #2, which helped them move into the preparation stage by creating a quit plan. In Week 3, participants went through Workbook #3, which helped them initiate the action stage. In Week 4, participants completed Workbook #4, which provided guidance on maintenance of the behavior change and ways to prevent the relapse stage.

Information and skills provided in the workbooks were designed based on Islamic and Arabic cultural values and assumptions [13]: deep religious orientation, reliance on the extended family, defined gender roles and taboos, use of the Arabic language, and adherence to traditional beliefs and practices. Social and family concerns, rather than a focus on the individual, were also emphasized in the intervention.

Reviews of telephone and face-to-face counseling sessions demonstrated that these interventions can be equally effective in enhancing quit rates [14,15]. This study included four individual 30 min telephone counseling sessions. Phone counseling content was primarily drawn from MI techniques to increase a participant’s motivation to change. Session topics included coping with withdrawal, maintaining a commitment to continued abstinence, and relapse prevention. Each participant was assigned to the same counselor in order to optimize rapport and familiarity with the participant’s unique concerns and smoking history.

Recruitment and initial data collection occurred at places of faith and community centers in the North and South Carolina area, which were frequented by ArAs. Follow-up counseling sessions were provided over the phone. Questionnaires were completed by participants online via the link provided in an email. Carbon monoxide levels were measured by participants at home and shared with the investigators via email.

We used a non-random, convenience sample of self-identified ArAs from faith and community centers in the metropolitan Charlotte, NC, and Fort Mill, SC areas. With high attendance in places of recruitment, high smoking prevalence among ArA, and ArA’s high interest in culturally tailored smoking cessation help (based on prior studies) [16,17], we anticipated great recruitment possibilities.

Recruitment flyers were posted on information boards of the selected faith and community centers. Potential participants were directed to the study investigators. Interested participants were told about the study and given an opportunity to provide informed consent (if interested). After consent, verification of tobacco smoking was completed with carbon monoxide monitors. This also provided a baseline carbon monoxide marker. Participants were also asked to complete a contact form and a series of questionnaires.

Potential participants had to (a) be adults (18–65 years of age); (b) self-identify as ArA; (c) have smoked or puffed on a cigarette during the previous week; (d) have an exhaled carbon monoxide (Ex CO) reading of 10 ppm or higher as an indicator of current smoking; (e) be able to read, write, and speak Arabic; (f) be willing to participate in the smoking cessation study; (g) have telephone and internet access; (h) be present in the geographical study area for 3 months; (i) not be currently enrolled in another smoking cessation treatment program. The fifth criterion was not restrictive, given that the recruitment settings offer their services in the Arabic language.

Potential participants were excluded if they had a plan to move out of North Carolina or South Carolina within 3 months following study enrollment. Participants who (a) had a myocardial infarction in the last 2 weeks, or (b) had been hospitalized for a heart-related condition in the prior 2 weeks, were allowed to participate in the study but did not receive nicotine replacement therapy due to contraindications.

The measurement instruments that were used in this study have been tested and validated [1,16,17,18] in ArA communities. With the exception of the “Demographics, c smoking history” form, which was administered only at recruitment (Baseline/Week 0), all other questionnaires were completed at baseline (Week 0), Week 5, and Month 3.

A demographics, smoking history form was used to obtain background and cultural information. The form included a 7-item Arab Acculturation Scale, a measure of smoking history, smokeless tobacco use, smoking habits, past attempts to quit smoking, and the desire to quit.

The Fagerström Test for Nicotine Dependence [19,20] is a 6-item scale that was used to measure the level of nicotine dependence or addiction. It assesses how soon tobacco use begins each day, the number of cigarettes during the day a person could do without, how smokers cope in places where they cannot smoke, and how frequently and how deeply they smoke.

The Perceived Self-Efficacy/Temptation Scale [21,22,23] is a 9-item self-efficacy scale commonly used to measure a participant’s confidence in his ability to abstain from cigarette smoking in a variety of different situations.

The Minnesota Withdrawal Scale [24] is a 15-item scale that measures smoking withdrawal symptoms, including anger, anxiety, cravings, depression, difficulty concentrating, hunger, impatience, insomnia, and restlessness.

Smoking cessation and reduction were assessed by asking participants whether they had smoked or had a puff on a cigarette in the past 7 days, and the number of cigarettes smoked in the past 7 days.

To assess a biomarker of smoking status, a Carbon Monoxide biochemical marker was used to validate self-reported smoking cessation and reduction. Carbon monoxide (CO) concentration in parts per million (ppm) from expired breath was measured using the Smokerlyzer meter. An Ex CO reading of 10 ppm or higher was used as an indicator of current CO exposure and smoking [25,26].

Due to a low participation number, researchers did a qualitative review of responses during the phone counseling session and summarized the responses.

The faith and community centers in which the recruitment and baseline measurements occurred allowed investigators to post recruitment fliers on announcement boards and provide separate activity rooms to ensure participants’ privacy and confidentiality during the baseline meeting. Follow-up counseling performed over the phone, provision of personal CO monitors, and completion of questionnaires online via the provided link increased convenience to the already busy participants. Location of the recruitment areas ensured success as each was selected based on a higher population of ArAs in metropolitan areas and its proximity to investigators’ residences.

Recruitment took place between January 2022 and May 2022. Advertisements and flyers were distributed at Middle Eastern grocery stores, restaurants, lounges, and faith-based organizations in Charlotte and Wilmington, inviting ArA to participate in a telephone counseling smoking cessation study. These materials included the Arabic-speaking co-investigator’s contact number for those interested in joining. Potential participants who responded to the advertisements were identified through a brief screening process. Data collection took place at mosques, churches, cultural centers frequented by ArA, as well as at the Middle Eastern establishments. During the screening process, conducted either by phone or in person, potential participants were referred to the Arabic-speaking research team members. Interested potential participants were invited to attend a screening session, during which they were informed about the study and given the opportunity to provide consent if they chose to participate. After consenting, participants answered a series of screening questions to determine their eligibility and provided demographic information. They also completed a contact form and a set of questionnaires.

Out of thirty potential participants, only twelve met the above-mentioned eligibility criteria. Incentives for completing assessments included USD 10 gift card for each follow-up. Participants could withdraw from the study at any time by informing the research team via telephone or in writing. Table 1 describes the measurement instruments utilized in the study.

The intervention consisted of four individual Telephone Counseling (TC) sessions for a period of 30 min each over an 8-week period. (See Figure 1 for the TC protocol). TC sessions were conducted by the study investigators, who are certified in MI training, and the content was primarily drawn from MI techniques to increase a participant’s motivation to change. There were two people who recruited and followed up with the participants. One of them was female and one was male. Both therapists spoke fluent Arabic. Session topics included coping with withdrawal, maintaining a commitment to continued abstinence, and relapse prevention.

## 3. Results

Of the 12 participants, only four completed the study. The study results are presented in Table 2, below. All participants received the same interventions: telephone MI counseling sessions and stress reduction (no complete cessation).

### 3.1. Analysis of Key Observations and Responses: Common Themes

#### 3.1.1. Anxiety and Stress

All participants expressed significant anxiety and stress related to delaying their first-morning cigarette. This suggests that the habitual nature of morning smoking might be a strong psychological barrier to cessation.Managing stress and anxiety could be a central focus of the intervention, possibly through relaxation techniques or adjusting the approach to reducing morning cravings.

#### 3.1.2. Relapse

All cases experienced relapse, which is common in smoking cessation. The challenge of complete cessation is evident, as participants reduced consumption but could not quit entirely.

#### 3.1.3. Fagerström Test Scores

The Fagerström Test for Nicotine Dependence is a helpful indicator of addiction severity, and the scores indicate varying levels of dependence. While most participants showed a slight reduction in their scores, full cessation did not occur, even in individuals with high nicotine dependence (score 6/6).

#### 3.1.4. Telephone MI Counseling

MI over the phone seemed to help reduce anxiety and stress, but it was not enough to achieve complete cessation. This could point to the need for a more tailored approach, perhaps focusing on building self-efficacy, addressing triggers, and offering more intensive support.

#### 3.1.5. Minnesota Nicotine Withdrawal Scale

Lower withdrawal symptoms in some participants (e.g., Cases 1, 2, and 3) may indicate that nicotine dependence is not the only factor contributing to relapse. Case 4, however, showed high withdrawal symptoms, suggesting that physical cravings could have been a more significant barrier for this participant.

#### 3.1.6. Fear of Counseling

A key issue for some participants was fear regarding the counseling calls, especially in the first weeks. This could indicate a need for more reassurance and more explicit expectations regarding the process of cessation counseling.

### 3.2. Recommendations for Improvement

#### 3.2.1. Addressing Anxiety and Stress

Specific strategies targeting morning cravings and anxiety could improve outcomes. This might include mindfulness practices, behavioral strategies to delay the first cigarette, or exploring other ways to manage stress (e.g., exercise, breathing exercises).

#### 3.2.2. Counseling Approach

To overcome the fear of counseling, it may be helpful to provide more detailed information about what the sessions will involve, ensuring participants understand the collaborative nature of MI and that it is not a judgmental process.

#### 3.2.3. Follow-Up and Continued Support

The telephone MI counseling sessions seemed to help reduce stress, but were not sufficient for complete cessation. A more long-term approach, with more frequent check-ins or additional resources (e.g., nicotine replacement therapy), could be more effective in achieving full smoking cessation.

#### 3.2.4. Customization of the Approach

Tailoring the counseling approach based on the individual’s level of nicotine dependence (Fagerström score) and specific anxiety and stress triggers could lead to more effective outcomes. For example, Case 4 (with higher withdrawal symptoms) might benefit from a more intense focus on managing cravings.

## 4. Discussion

This case study provides an assessment of the need for telephone counseling (TC) as a smoking cessation strategy within Arab American (ArA) communities, particularly in light of the high tobacco prevalence often linked to immigration from politically unstable regions of the Middle East. The findings demonstrate a possibility of tailoring TC services to meet the unique challenges faced by these communities, including cultural pressures, social norms, and elevated rates of tobacco use. Additionally, the integration of Nicotine Replacement Therapy (NRT) with TC has the potential to further enhance cessation outcomes.

Only a handful of studies have focused on ArA’s tobacco use trends in the past 15 years, and very few of these provided culturally tailored interventions [1,9,27]. Although Arab Americans have a high rate of tobacco use, they continue to have limited access to culturally tailored tobacco cessation services in their communities [27]. One of the ways to improve this access is to use telehealth (e.g., telephone counseling). The current study demonstrates the areas for improvement in this method, specifically, it highlights the challenges in engaging and keeping ArAs engaged in the intervention and follow-ups. While we had success with 60% phone counseling retention and remote follow-ups in our prior studies [1,10] in Florida and Virginia, participants in this study were less engaged. This may be explained by the added stressors of the COVID-19 pandemic, the volume of information collected, fatigue from remote communication (preference for live interactions), or lower interest in quitting during the time of study. Additionally, the challenges in recruitment and retention in our current study may be attributed to the participants’ location (e.g., Wilmington is a smaller town that has lower diversity and a higher likelihood of recognition of participants’ responses). Future studies will need to explore these challenges further by studying a bigger sample of participants.

Although this exploratory study did not yield statistically significant quit rates, it offered valuable insights into the necessity of culturally appropriate services. The inclusion of four individual cases lends credibility to the intervention’s relevance, even if broader statistical significance was not achieved.

The findings emphasize the importance of addressing issues specific to the Arab American experience, such as family pressures and cultural expectations surrounding smoking. However, the intervention fell short in fully incorporating these cultural aspects. Furthermore, participants’ requests for e-cigarette cessation support indicate that existing smoking cessation programs may need to broaden their scope to address newer forms of nicotine use, such as vaping. The TC intervention could benefit from a deeper integration of cultural competencies, particularly regarding social and family dynamics that influence smoking habits within Arab American communities. Providing culturally specific resources and developing problem-solving techniques that address the influence of family and community dynamics could significantly enhance participant engagement and the program’s effectiveness. The study acknowledges several limitations inherent to its case study design. The findings are not easily generalizable to a larger population, and more robust trials are needed to establish broader applicability. Additionally, the absence of long-term follow-up is a critical limitation, as smoking cessation—especially among populations with high tobacco use—requires sustained support and tracking to assess the intervention’s true effectiveness.

Some participants expressed the need for support with e-cigarettes, highlighting the importance of adapting TC to address both traditional tobacco use and newer nicotine delivery systems like vaping. Incorporating these considerations into the intervention could make it more comprehensive and relevant to participants’ needs. Given these limitations, a larger-scale study employing a randomized controlled trial design is strongly recommended to assess the effectiveness of TC, particularly when conducted in Arabic. Collecting long-term follow-up data is also crucial to provide a fuller understanding of the intervention’s success over time.

Future interventions should address the cultural nuances specific to Arab American communities and broaden the scope of cessation programs to include support for alternative nicotine products. Strategies such as peer support and collaboration with community health programs could further enhance program outcomes.


**Limitation**
**:**


This quasi-experimental study employed convenience sampling without a control group, which may have introduced selection and sampling biases and reduced internal validity. Although three investigators followed the same protocol for data collection, participants’ perceptions could have contributed to gender, authority, or experimenter biases. The small sample size and participant attrition further limited both internal and external validity. Additionally, cultural factors may have influenced the findings. In many Arab contexts, personal connections and face-to-face interactions are highly valued, and phone interviews can create anxiety or discomfort. This cultural preference may have affected participants’ engagement and the depth of their responses.

## 5. Conclusions

This study highlights that culturally tailored telephone counseling has weak potential to be an effective smoking cessation strategy for Arab Americans, though further research and refinement are essential. By addressing cultural factors, expanding services to cover emerging nicotine products, and scaling up research efforts, this approach could become a valuable tool in tobacco control efforts for this demographic. Future research should focus on quantifying outcomes, integrating long-term follow-up, and exploring additional interventions to create a more comprehensive and impactful cessation program.

## Figures and Tables

**Figure 1 ijerph-22-01453-f001:**
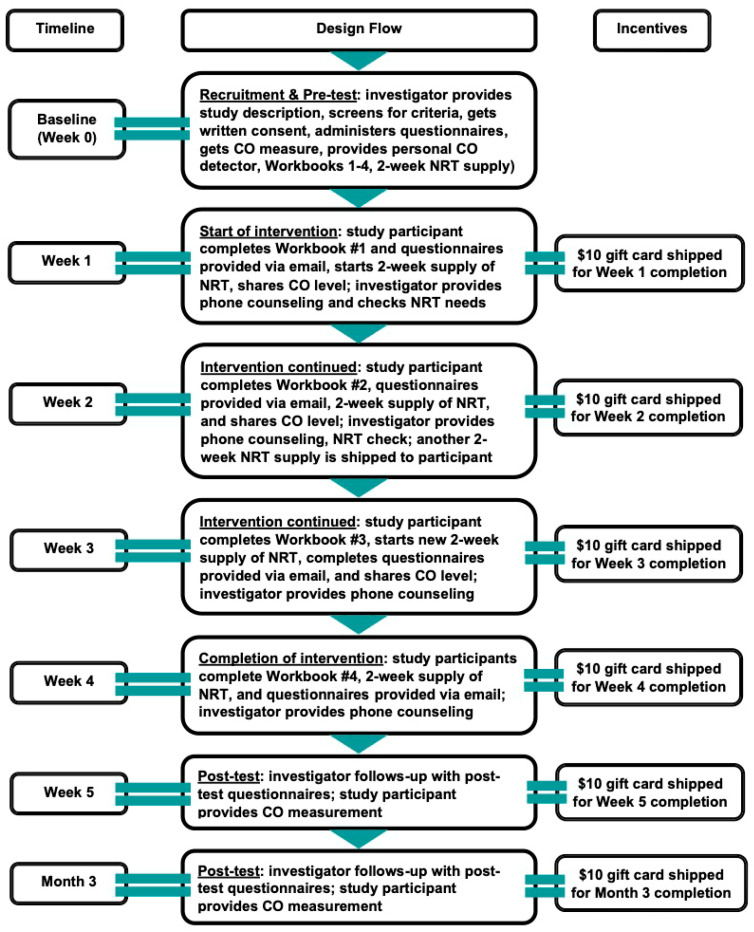
Design of the study.

**Figure 2 ijerph-22-01453-f002:**
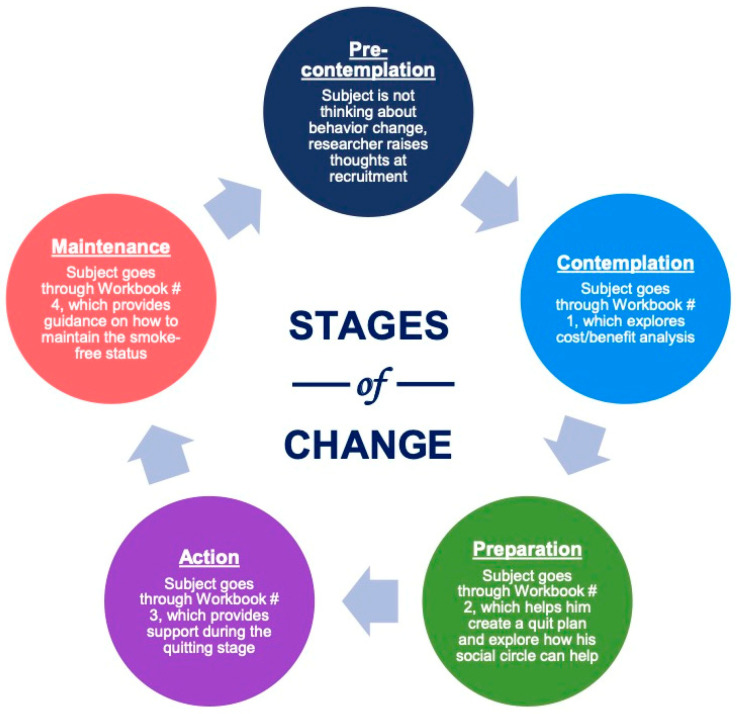
Application of the transtheoretical model (TTM).

**Table 1 ijerph-22-01453-t001:** Measurement Instruments.

Instrument	Description
Smoking reduction and cessation	The number of cigarettes and/or puffs smoked within the past 7 days.
Fagerstrom Test for Nicotine [19,20]	A standard instrument for assessing intensity of physical addiction to nicotine. The test was designed to provide an ordinal measure of nicotine dependence related to cigarette smoking.
The Perceived Self-Efficacy/Temptation Scale [21,22,23]	This scale is a 9-item self-efficacy scale commonly used to measure a participant’s confidence in his ability to abstain from cigarette smoking in a variety of different situations.
Exhaled Carbon Monoxide Test [25,26]	Carbon Monoxide biochemical marker was used to validate self-reported smoking cessation and reduction. Carbon monoxide (CO) concentration in parts per million (ppm) from expired breath will be measured using the Smokerlyzer meter.
Minnesota Tobacco Withdrawal Scale [24]	Features two separate measures for examining the severity of nicotine withdrawal symptoms in a subject: a self-report scale and an observer scale. The observer scale asks the scale-taker to rate the severity of four symptoms in someone they know who is experiencing nicotine withdrawal: “angry/irritable/frustrated,” “anxious/tense,” “depressed,” and “restless/impatient.” The self-report version asks for rankings of the severity of those same four symptoms, plus eleven others that cannot be observed by outsiders (including things as “desire or craving to smoke,” “insomnia, sleep problems, awakening at night,” or “dizziness”). Both scales use a Likert-type scale for the severity ratings, ranging from 0 (“not at all”) to 4 (“severe”).

**Table 2 ijerph-22-01453-t002:** Smoking Cessation and Reduction.

Case Number	Age	Cigarettes per Day	Ex CO (PMM)	Fagerstrom Test	MNWS	Challenges	Key Observations and Responses
1	65	10 10 after 28 days	10 10 after 28 days	6/6 (Baseline) 5/6 (after 28 days)	Low	1. High anxiety/stress if postponing first-morning cigarette. 2. Relapse. 3. Fear of counseling in first 4 weeks.	1. Reduction in cigarettes was achieved, but quitting was not successful. 2. Anxiety during morning delayed adherence. 3. Fear of early counseling calls reduced engagement.
2	58	10–15 10–12 after 28 days	12 12 after 28 days	5/6 (Baseline) 4/6 (after 28 days)	Low	1. Anxiety and Stress from delaying the first-morning cigarette. 2. Relapse. 3. Fear Related to counseling calls in the first 2 weeks.	1. Reduction in cigarettes was achieved, but quitting was not successful. 2. Anxiety related to morning cigarette delay was a barrier to progress. 3. Participant's fear of counseling calls may have interfered with his engagement in the process.
3	28	10 10 (after 28 days)	Unknown (presumably 10 based on similar cases)	6/6 (Baseline) 5/6 (after 28 days)	Low	1. Anxiety and Stress from delaying the first-morning cigarette. 2. Relapse.	1. Reduction in cigarettes was achieved, but quitting was not successful. 2. Anxiety related to morning cigarette delay was a barrier to progress. 3. There was no indication of significant difference in counseling engagement compared to the other cases.
4	30	15–20 15–20 (after 28 days)	Missing 10 (after 28 days)	6/6 (Baseline) 5/6 (after 28 days)	High	Anxiety and Stress from delaying the first-morning cigarette.	1. Reduction in cigarettes was achieved, but quitting was not successful. 2. High withdrawal symptoms may have made cessation more difficult. 3. Similar anxiety issues to other participants, possibly contributing to more difficulty in quitting.

## Data Availability

Baseline and repeated survey data can be found, respectively, at https://uncw.az1.qualtrics.com/jfe/form/SV_3kOqNeq8laGLF9c. https://uncw.az1.qualtrics.com/jfe/form/SV_0lEP7LHtkfXbpQi. Accessed on 18 May 2025.

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
