# Peer review of "Replication of a Culturally Tailored Tobacco Cessation Intervention for Arab American Men in North Carolina: An Exploratory Pilot Study"

_ijerph, 2025, doi:10.3390/ijerph22091453_

Round 1

Reviewer 1 Report

Comments and Suggestions for Authors

Interesting study considering tailoring intervention for smoking cessation for a specific male population.  I have a few queries.

  • why’s was the intervention so limited given we know  this is not best practice eg no NRT / varenicline. 4 weeks duration too brief . While we need to tailor inventions we cannot have baseline was less than usual.

    - please provide rationale and ethics to establish the study   Please also add to limitations section
  • - add limitations section
  • harm reduction e cigs are not NRT globally- whole body g evidence that harm reduction are words used by tobacco industry to sell other products. 

Author Response

  • why’s was the intervention so limited given we know  this is not best practice eg no NRT / varenicline. 4 weeks duration too brief . While we need to tailor inventions we cannot have baseline was less than usual.
    • Rationale: The 4-week intervention is what we could afford based on our limited funds and it replicates the structure that we had in the Florida study (which did have successful results among ArA men).

      - please provide rationale and ethics to establish the study   Please also add to limitations section
  • - add limitations section
    • Thank you for asking to add a limitation section; we will do that, and here is what we going to add:  "This is a quasi-experimental study with convenience sampling and no control group, which can create bias (e.g., selection and sampling biases) and reduce internal validity. Data was collected by three different investigators, who followed the same protocol. However, depending on the participants’ perception, there is a possibility for gender, authority, and experimenter biases. Small sample size and attrition in this study may have further reduced internal and external validities."
    • We Also added Limitation section
  • harm reduction e cigs are not NRT globally- whole body g evidence that harm reduction are words used by tobacco industry to sell other products. 
    • Thank you, this was addressed in the document.

Reviewer 2 Report

Comments and Suggestions for Authors

I thought the paper was well organized and clearly written. I also appreciated the application of the Transtheoretical Model and the overall approach. Below are my comments and suggestions:

  • Sample size: The overall sample size is very small. Only 30 potential participants were recruited, compared to 180 in the Florida study referenced. Given that only a fraction of participants typically complete such studies (as seen in Florida study), it would have been advisable to continue recruitment beyond 30. This appears to be a significant oversight in the study design. While the authors acknowledge in the introduction that "larger trials are needed to confirm its effectiveness," they did not attempt to recruit a sufficiently large sample to support even preliminary quantitative assessment. 
  • Participant Characteristics: One of the inclusion criteria was having smoked a cigarette in the past 7 days. However, it's unclear whether participants were established or exclusive cigarette smokers, or if they were also using other tobacco products. Clarifying this would help contextualize the findings. 
  • Cultural Considerations: The use of counseling and relaxation techniques may not be widely accepted or culturally appropriate in Arab contexts and could potentially increase anxiety for participants. This warrants further discussion.
  • Misleading Citations (Lines 40-41): The statement that "In many Middle Eastern countries, smoking is a common practice among men, symbolizing hospitality and maturity (Rice et al. 2006), with prevalence rates reaching 69% among men (Cornelius et al. 2020 and Ghadban et al. 2019)" is not supported by the cited sources.
    • Rice et al. (2006) focuses on predictors of tobacco use among Arab American adolescents - not adult men in Middle Eastern countries. 
    • Cornelius et al. (2020) is an MMWR based on NHIS data, which pertains to the US population.
    • Ghadban et al. examines smoking behaviors and cessation motivations among Arab Americans, not prevalence in Middle Eastern countries. 
    • Additionally, the cited prevalence rate of 69% among men appears to be inaccurate and should be corrected and properly sourced.

Author Response

We thank you for the feedback and below in red are our responses to your feedback.

  • Sample size: The overall sample size is very small. Only 30 potential participants were recruited, compared to 180 in the Florida study referenced. Given that only a fraction of participants typically complete such studies (as seen in Florida study), it would have been advisable to continue recruitment beyond 30. This appears to be a significant oversight in the study design. While the authors acknowledge in the introduction that "larger trials are needed to confirm its effectiveness," they did not attempt to recruit a sufficiently large sample to support even preliminary quantitative assessment. 
    • The sample size of 30 was based on the funding limitations and was aimed for the purpose of the pilot project. There were continuous attempts to recruit more participants during the pandemic years (2020-2022); however, there were not many people who were interested in participation. Thus, the study had to stop before reaching the aimed sample size (30) because the internal grants allowed funding for a limited time.
  • Participant Characteristics: One of the inclusion criteria was having smoked a cigarette in the past 7 days. However, it's unclear whether participants were established or exclusive cigarette smokers, or if they were also using other tobacco products. Clarifying this would help contextualize the findings. 
    • To maximize recruitment, we were open to dual/multiple tobacco users and exclusive smokers. We added that into the manuscript.
  • Cultural Considerations: The use of counseling and relaxation techniques may not be widely accepted or culturally appropriate in Arab contexts and could potentially increase anxiety for participants. This warrants further discussion.
    • We added that into the Discussion
  • Misleading Citations (Lines 40-41): The statement that "In many Middle Eastern countries, smoking is a common practice among men, symbolizing hospitality and maturity (Rice et al. 2006), with prevalence rates reaching 69% among men (Cornelius et al. 2020 and Ghadban et al. 2019)" is not supported by the cited sources.

All were corrected and updated.

Reviewer 3 Report

Comments and Suggestions for Authors

This is a well-written manuscript on a high-risk smoking population, that has received little attention except by the same group of authors in a previous state. The topic deserves greater attention, and the authors have developed a reputable pilot intervention.

The manuscript could benefit from some additional attention to details.

  1. The abstract methodology is very sparse in details. I was surprised in further reading that the intervention involved NRT and CO measurements, and that the duration of the intervention was actually somewhat extensive. Please add this information to the abstract methods. If need of space, the abstract conclusion can be shortened to just its first sentence.
  2. In the discussion section, state the completion rates for Virginia and Florida so the reader has an understanding of the differences in NC. 
  3. It is stated that there was a qualitative review of responses. Please provide more details. Is the qualitative review the responses listed in Table 2 under the column "Challenges?"
  4. It's unclear why the respondents discussed delaying the first cigarette. Why would this come up in conversation, unless they were prompted with this question. But it is also unclear why this would be a question, unless delaying the first cigarette is part of the intervention. It doesn't seem like part of the intervention.
  5. The paper is missing some important information that is relevant to the feasibility of the intervention. Namely the reasons for drop-out. The authors must have collected this information. Similarly, attrition might have varied by FTND, CO or any number of other factors. This should be discussed. Another important factor in attrition is when the subjects dropped out. Was it early on in the study or after their NRT course was finished? Even though the sample size of drop-outs is small, it's larger than that for the completers. 
  6. I assume the CO and CPD shown in Table 2 was baseline. What was it at the end of the study? Can that be put in Table 2 as well. 
  7. It's clear that the authors wish to promote the cessation program and that is understandable. I don't know that it really shows promise as concluded by the abstract. Few participants completed the program, none quit, and the FTND was basically unchanged among those who completed.
  8. If it shows promise then this conclusion is more likely supported by the results in all 3 states than NC, but then the authors need to make the case. As it is, there is little discussion about the findings from other states. 

Author Response

We want to thank you for your feedback and suggestions. We have addressed the different suggestions pointed below in red, addressing every point that was suggested.

1. The abstract methodology is very sparse in details. I was surprised in further reading that the intervention involved NRT and CO measurements, and that the duration of the intervention was actually somewhat extensive. Please add this information to the abstract methods. If need of space, the abstract conclusion can be shortened to just its first sentence.

Thank you, we added that into the abstract.

2. In the discussion section, state the completion rates for Virginia and Florida so the reader has an understanding of the differences in NC. 

We  added that in the script which was about 60%.

3. It is stated that there was a qualitative review of responses. Please provide more details. Is the qualitative review the responses listed in Table 2 under the column "Challenges?"

We addressed the details of information in table 2.

4. It's unclear why the respondents discussed delaying the first cigarette. Why would this come up in conversation, unless they were prompted with this question. But it is also unclear why this would be a question, unless delaying the first cigarette is part of the intervention. It doesn't seem like part of the intervention.

There is a question in both the baseline and the follow-up surveys that asks them how soon after waking up they need to have that first cigarette. This is done with the purpose of deciding whether they need 2 or 4 mg, which is based on what’s recommended on the back of the nicotine lozenges/gum boxes.

5. The paper is missing some important information that is relevant to the feasibility of the intervention. Namely the reasons for drop-out. The authors must have collected this information. Similarly, attrition might have varied by FTND, CO or any number of other factors. This should be discussed. Another important factor in attrition is when the subjects dropped out. Was it early on in the study or after their NRT course was finished? Even though the sample size of drop-outs is small, it's larger than that for the completers. 

Attrition occurred after 4 weeks, most likely due to cultural reasons.

6. I assume the CO and CPD shown in Table 2 was baseline. What was it at the end of the study? Can that be put in Table 2 as well. 

We have added that content into the manuscript.

7. It's clear that the authors wish to promote the cessation program and that is understandable. I don't know that it really shows promise as concluded by the abstract. Few participants completed the program, none quit, and the FTND was basically unchanged among those who completed.

Correct, the study did not show any promise and we changed that in the abstract

8. If it shows promise then this conclusion is more likely supported by the results in all 3 states than NC, but then the authors need to make the case. As it is, there is little discussion about the findings from other states. 

Thank you, we changed the conclusion and we added an explanation of that part of  poor results. To maximize recruitment, we were open to dual/multiple tobacco users and exclusive smokers.

Round 2

Reviewer 2 Report

Comments and Suggestions for Authors

Thank you for clarifying and addressing my comments.

Author Response

We want to thank you for the time you spent to read our manuscript and provide your suggestions so we can provide the enhanced version of the manuscript.

Reviewer 3 Report

Comments and Suggestions for Authors

The manuscript is much improved. In responding to the reviewer comments, the authors have actually over-compensated. 

The abstract should be slightly modified to say that:

The study suggests that culturally-tailored telephone counseling did not show promise as a
smoking cessation strategy for Arab Americans in North Carolina due to low participation. 

The sample size is really too small to test the efficacy of the intervention itself. It seems to have been more successful in another state.

Please check references. In the conclusion for example, authors reference their previous study as reference 10. It is reference 9. 

Author Response

We want to thank you for your suggestions and comments as well as the attention to details to enhance the manuscript  

The manuscript is much improved. In responding to the reviewer comments, the authors have actually over-compensated. 

The abstract should be slightly modified to say that:

The study suggests that culturally-tailored telephone counseling did not show promise as a
smoking cessation strategy for Arab Americans in North Carolina due to low participation. 

The sample size is really too small to test the efficacy of the intervention itself. It seems to have been more successful in another state.

This was added to the abstract. Thank you

Please check references. In the conclusion for example, authors reference their previous study as reference 10. It is reference 9. 

This was corrected in the manuscript. Thank you